# Elaboration of Conductive Hydrogels by 3D Printer for the Development of Strain Sensors

**DOI:** 10.3390/gels11070474

**Published:** 2025-06-20

**Authors:** Lucas Carravero Costa, Isabelle Pochard, Cédric C. Buron, Florian E. Jurin

**Affiliations:** Université Marie et Louis Pasteur (UMLP), Institut UTINAM, UMR 6213 CNRS, F-25000 Besançon Cedex, France; lucas.carravero@gmail.com (L.C.C.); isabelle.pochard@univ-fcomte.fr (I.P.); cedric.buron@univ-fcomte.fr (C.C.B.)

**Keywords:** hydrogel, strain sensor, rheology, piezoelectricity, 3D printing, hybrid materials

## Abstract

The development of biocompatible, conductive hydrogels via direct ink writing (DIW) has gained increasing attention for strain sensor applications. In this work, a hydrogel matrix composed of polyvinyl alcohol (PVA) and κ-carrageenan (KC) was formulated and enhanced with polyvinylidene fluoride (PVDF) and silver nanoparticles (AgNPs) to impart piezoelectric properties. The ink formulation was optimized to achieve shear-thinning and thixotropic recovery behavior, ensuring printability through extrusion-based 3D printing. The resulting hydrogels exhibited high water uptake (~280–300%) and retained mechanical integrity. Rheological assessments showed that increasing PVDF content improved stiffness without compromising printability. Electrical characterization demonstrated that AgNPs were essential for generating piezoelectric signals under mechanical stress, as PVDF alone was insufficient. While AgNPs did not significantly alter the crystalline phase distribution of PVDF, they enhanced conductivity and signal responsiveness. XRD and SEM-EDX analyses confirmed the presence and uneven distribution of AgNPs within the hydrogel. The optimized ink formulation (5% PVA, 0.94% KC, 6% PVDF) enabled the successful fabrication of functional sensors, highlighting the material’s strong potential for use in wearable or biomedical strain-sensing applications.

## 1. Introduction

Improving the devices used in the biomedical field is necessary to have more data regarding human activities and in this way produce more efficient apparatus for human detection motion and healthcare monitoring [1,2,3].

Usually, the classical strain sensors used for healthcare applications have problems with biocompatibility or poor mechanical properties such as high stiffness due to the components used in their manufacturing process [4].

One approach to overcome the usual problems is the use of hydrogels as strain sensors [1,5,6]. Hydrogels are three-dimensional polymeric networks that are able to uptake water from the surroundings and swell, without losing mass during this process [5]. Due to their viscoelastic properties, biocompatibility, and good mechanical properties, these materials are used for many applications, such as drug delivery or cell culture [6,7,8]. Depending on the components used in the hydrogel formulation, it can have chemical or physical crosslinks between the polymeric chains [9,10]. Therefore, to elaborate sensors from hydrogels, there is a need to convert these materials into electrically responsive systems, as usually these networks do not intrinsically have this characteristic.

Regarding the production of the final product, molding (traditional) or additive manufacturing techniques can be used [11]. Despite the advantages of the traditional approaches, 3D printing permits the fabrication of materials with complex structures that can be very useful for personalized applications by deposing layer on layer [2,5].

The selection of the composite hydrogel components was guided by the objective of developing a biocompatible, printable, and piezoelectrically responsive material suitable for strain-sensing applications. For the creation of the polymeric matrix, polyvinyl alcohol (PVA) and κ-Carrageenan (KC) were selected. Both components are biocompatible and can form hydrogels due to intermolecular interactions (van der Waals and hydrogen bonds) [6,7,12]. Moreover, it was already described in the literature that the mixture between these two polymers can be used as an ink for 3D printing via direct ink writing (DIW), as it presented adequate rheological properties for this purpose [6].

To provide the electrical response to the material, polyvinyl fluoride (PVDF) and silver nanoparticles (AgNPs) were added to the formulation. PVDF is a polymer that has already been used for biomedical applications, such as in the preparation of wound-healing devices, it is easy to extrude, and depending on the crystalline phase (β-phase) present it gives piezoelectricity, which makes a self-supply device, avoiding the use of external batteries and making the design easier [13]. AgNPs were added to the final product formulation, as they can be prepared easily, have already been used in biomedical applications, and can improve the piezoelectrical properties of PVDF [13,14].

Concerning the method for 3D printing, due to the materials selected, DIW was the chosen method. In this technique, the ink is extruded by the print head, so knowing the rheological behavior of the ink is extremely relevant [6,15].

Thus, this project aimed to design and fabricate an electrically sensitive and wearable hydrogel using 3D printing for strain sensing applications for instance in the healthcare sector.

This study aimed to design and to fabricate an electrically sensitive hydrogel using 3D printing technology for use in strain-sensing applications within the healthcare sector. The overall research and development process was divided into three principal phases. The first phase focused on the development and characterization of the ink formulation. This involved optimizing the ratios of PVA, KC, PVDF, and AgNPs to achieve a composition with appropriate rheological, mechanical, and electrical properties. Detailed analyses, such as viscosity profiling and oscillatory rheometry, were performed to ensure that the ink would perform well under DIW conditions. The second phase involved the characterization of the structured hydrogels fabricated using the optimized ink. This step included evaluating the mechanical properties (e.g., elasticity, tensile strength), water content, and swelling behavior. Additionally, the electrical performance of the hydrogels was assessed, with particular attention given to their piezoelectric response under cyclic strain. The final phase addressed the development of a robust 3D printing protocol tailored to the prepared ink. The protocol was iteratively refined to ensure consistency and reproducibility in the fabrication of strain sensors.

## 2. Results and Discussion

### 2.1. Efficiency of the DMF Removal Step

DMF was used as a cosolvent in the ink formulation due to its effectiveness in dissolving polyvinylidene fluoride (PVDF), a hydrophobic polymer that does not dissolve efficiently in water alone. During the ink preparation, the water-DMF mixture (7:3 *v*/*v*) allowed for homogeneous dispersion of the PVDF in the hydrogel matrix during preparation, thus ensuring good PVDF distribution.

Considering the final application of the product, it was important that the N,N-dimethylformamide (DMF) was removed from the structured hydrogels. It was desired that this compound be removed from the final product, as it can be toxic for human beings [16]. Figure 1 shows IR spectra collected from pure DMF, a hydrogel prior to the DMF removal, and a hydrogel after DMF removal.

Among all the compounds used in the ink formulation, DMF is the only one that contains a C=O bond. Therefore, the stretching of this bond, which is an active mode in the region between 1665 and 1760 cm^−1^ with a particular shape was used to detect the presence of this material.

The peak at approximately 1662 cm^−1^ refers to the carbonyl group. So, the rinsing step was validated, as the C=O stretching band is not present in the spectrum of the sample that should not have DMF (red curve). Regarding the manufacturing process of the final product, the effectiveness of the step to remove DMF was verified.

### 2.2. Rheological Assessment of Inks

In order to study the impact of PVDF concentration in the ink formulation, different rheological characterizations were carried out.

#### 2.2.1. Flow Test Curves

Figure 2a shows the viscosity measurement performed with the inks at different PVDF concentrations.

All the formulations presented a shear-thinning behavior, as their viscosity is higher at lower shear rates and lower at higher shear rates. The concentration of PVDF inside the ink formulation does not have a major influence on the final viscosity of the inks. This shear-thinning behavior was mainly due to KC and PVA present in the formulation, as already observed for similar hydrogels [6].

#### 2.2.2. Amplitude and Frequency Sweep Measurements

Figure 2b,c show amplitude and frequency sweep performed with the inks at different PVDF concentrations.

As shown in Figure 2b, all the materials had a similar linear viscoelastic region (around 50%), which proved that PVDF which proved that PVDF does not allow the material to be deformed further. This was expected, as PVDF is a hydrophobic material and the polymeric matrix is hydrophilic, so there are not strong intermolecular interactions such as hydrogen bonds between PVDF and the other polymers that would increase the strength of the material due to the formation of new crosslinks. However, as more PVDF was added, there was an increase in G’, showing that increasing PVDF content made the material more rigid. This is probably due to the higher presence of polymer in the system, which increases the number of interactions between the materials by van der Waals interactions, making it more structured.

Frequency sweep measurements shown in Figure 2c agreed with the data shown for the amplitude sweep experiment (Figure 2b), in which the higher concentration of PVDF brought more rigidity. Moreover, the presented results are consistent with expectations for gel-like materials in the linear viscoelastic region where G′ is greater than G″ over the entire frequency range studied. Comparing the values of elastic and viscous moduli with the rheological values of biological tissues, it turns out that the inks have similar properties to reconstituted ECMs like Type-1 collagen and fibrin materials [17]. This is an interesting property for inks for biological applications such as enhanced cell development.

#### 2.2.3. Shear Stress Sweep Measurements

The results regarding the stress sweep experiment are given in Figure 2d.

As in the amplitude sweep experiment, the elastic modulus was higher than the viscous modulus for all the samples. Also, increasing PVDF concentration increased the elastic modulus values. This experiment was performed to identify the yield stress of each sample. The yield stress is the minimum value that the shear stress should have to make the material flow [7]. Thus, in the context of 3D printing by extrusion, the stress applied to the ink while printing must be higher than the yield stress so that the ink can flow [18].

With the yield stress values, it is possible to define the limit of the linear viscoelastic region in terms of stress. Additionally, it is possible to perform the thixotropic loop experiment in terms of stress, as the high value of stress applied should be beyond the yield stress (in the non-linear viscoelastic region).

The yield stress was determined graphically by the point that G′ and G″ cross each other. Based on the results shown in Figure 2d, the yield stress was approximately 170 Pa, 226 Pa, and 208 Pa for the inks PVDF6, PVDF8, and PVDF10, respectively. Based on that, for the thixotropy loop experiment, the high-stress value was set at 350 Pa.

Figure 3 shows the rheological properties below and above the yield stress to characterize the thixotropic properties of the inks. All the inks demonstrated a dominant solid-like behavior at low stress and a viscous-like behavior at high stress (outside the linear viscoelastic region). This agrees with the previous results. As the inks went from a more elastic to a more viscous-like behavior under the application of shear stress and could recover over time, it could be stated that these samples showed thixotropic recovery.

So, considering all the results from the rheological characterization performed for the inks with different concentrations of PVDF, it could be stated that this factor did not significantly affect the properties of the ink. To ensure its suitability for direct ink writing (DIW), the formulated inks were designed to exhibit key rheological properties, including shear-thinning behavior, thixotropic recovery, and adequate yield strength. These characteristics enable smooth extrusion and structural stability after deposition. Similar properties have already been obtained for PVA hydrogels [19,20].

### 2.3. Characterization of the Structured Hydrogels

#### 2.3.1. Amplitude Strain Sweep and Swelling Ratio Evaluation

With optimized composition, a swelling ratio assay was performed to verify the hydrogel profile of water absorption over time. The swelling ratio results are given in Figure 4a. Both hydrogels showed similar behavior, with higher absorption of water in the first 5 h and an average final swelling ratio in the range of 280–300% after 24 h. It can be stated that the silver nanoparticles did not interfere drastically with the water absorption of the hydrogel. The kinetic of the swelling ratio is quite slow compared to other PVA hydrogels already developed [21]. These can be due to the presence of PVDF, a hydrophobic polymer that can prevent the incorporation of water inside the materials. Low porosity can also explain slow water uptake, which is confirmed on the SEM images (Figures 8 and 9). No dissolution of the samples has been observed during the test.

Figure 4b presents the amplitude strain experiment results for the different structured hydrogels after the end of the manufacturing process. For most of the studied hydrogels, the samples with silver nanoparticles were more structured when compared with their respective pair without the nanoparticles. This result was expected, as it is already known that the inclusion of nanoparticles enhances the stiffness of the composites, as the metallic nanoparticles are intrinsically stiff [22].

#### 2.3.2. Piezoelectricity and Electrical Resistance Evaluation

The hydrogels without the nanoparticles did not generate a detectable piezoelectric response under compression, as with the generation of deformation due to the load applied, the voltage did not change. In contrast, the hydrogels with the silver nanoparticles showed a piezoelectric response, as shown in Figure 5.

The piezoelectric response from the hydrogels with 6% and 8% of PVDF was similar, with variations around 2.5–3 V, increasing their voltage under pressure. The hydrogel with 10% PVDF showed a smaller response, varying from approximately 1.5 to 3 V, despite having more content of the potential piezoelectric material. The piezoelectric results obtained are consistent with previous work on PVDF-based materials. An increase in the output voltage around 2 to 7 V could also be characterized as a function of an applied load [23,24]. Khazani et al. developed a piezoelectric material with PVDF and alginate with a voltage output signal ranging from 0.1 to 4 V for loads ranging from a few newtons to 40 N [25].

Comparing the samples with nanoparticles and without them, it can be assumed that just PVDF, KC, and PVA are not enough to provide detectable piezoelectricity. As KC and PVA are not piezoelectric materials, the only material that could bring piezoelectricity was PVDF. However, in the blank samples, PVDF alone was not enough to provide the electroactive response under mechanical stimulus.

Also tried was an evaluation of the piezoelectric performance of the hydrogel in the wet state. However, as shown with one example in Figure 5d, the response was much smaller than in the dry state. This reduction was probably due to the perturbations caused by the water molecules in the hydrogel, such as the formation of hydrogen bonds with the polymeric matrix, which could make the polarization in the hydrogel harder [26].

Another possibility for the decrease in the piezoelectric response is that water could turn the material more conductive due to ions conductivity, and this could interfere with the response [27]. To validate this hypothesis, resistance measurements were performed in the wet and dry state of the hydrogels.

Regarding the dry state, as expected the samples with nanoparticles showed a lower value of resistance when compared with the samples without them, as the silver nanoparticles are conductive materials [5]. The difference in resistance values is very significant between formulations with and without AgNPs. There are approximately 1.1 GΩ with AgNPs, while they range between 30 and 120 GΩ without particles. However, the resistance values of all samples were still high, in the range of GΩ, as shown in Table 1. This is due to the electrically insulating characteristics of PVA and PVDF [27,28]. Significant variability in resistance can be observed in the presence of AgNPs. The inhomogeneous distribution of AgNPs, creates areas of varying density in conductive particles, which locally modifies the percolation network and therefore the resistance values.

In the wet state, all the samples showed resistance values in the range of kΩ, proving that water affects the resistance of the samples, as it can generate ionic conductivity due to the ions present in the sample. These ions could be present in the raw materials, as counterions or impurities, as well as silver ions that were not reduced by ascorbic acid to form the nanoparticles. In addition, it was also verified that the samples with the AgNPs had a lower resistance. Regardless of the hydrogel state (dry or wet), the PVDF concentration has no impact, meaning that reducing the addition of insulating polymer does not improve the conductive properties. The percolation network required to achieve conductivity is not affected by the effect of these different PVDF concentrations.

#### 2.3.3. Evaluation of the Impact of AgNPs

One of the techniques that can be used to detect some PVDF polymorphs is IR spectroscopy. The absorption bands at 532, 613, 763, 795, 855, 976, 1186, and 1401 cm^−1^ belong to the α-phase. The absorption bands at 510, and 1278 cm^−1^ belong to the β-phase. The peaks at 485, and 879 cm^−1^ refer to the amorphous phase [13,29].

Figure 6a represents the IR spectra of a sample with nanoparticles, a sample without nanoparticles, and the pure PVDF used as raw material to prepare all the formulations.

In Figure 6a, it was possible to detect the α and β-phases of PVDF in all the samples analyzed by IR. Therefore, it can be stated that, through infrared analysis, it was not possible to detect a higher proportion of β-phase due to the presence of silver nanoparticles, as the ratio between the peaks referred to as the α and β-phases in each spectrum is kept constant.

To validate the hypothesis about the ratio between alpha and beta phases, XRD measurements were also performed between a sample with nanoparticles and a sample without them. The results are shown in Figure 6b. The peaks that characterize the α-phase are approximately 18.5° and 26.6°, while the peak that indicates the β-phase is at approximately 20° [29,30]. Thus, through XRD analysis, it could not be confirmed that the presence of the nanoparticles induced the formation of the piezoelectric phase, as the proportion between the peaks of alpha and beta phases was unmodified in both spectra.

The other peaks (38°, 44°, 64°, and 77°) describe the planes 111, 200, 220, and 311 of the silver nanoparticles, respectively [31]. As the AgNPs were present in the hydrogel, it was studied if they were homogeneously distributed in the hydrogel. To verify this information, XRD and SEM with EDX analysis were used.

Figure 7a presents XRD curves of the same sample but in three different regions (top, bottom, and middle).

Using the designation of the peaks mentioned before, it is possible to identify that the particles were not homogeneously distributed in the final product, as the AgNP peaks have higher intensity in one region than in the others. This information was also confirmed by the images generated by SEM using backscattering electrons (Figure 8), as the white parts (mainly indicated by the arrows in the image) represent the silver nanoparticles in the hydrogels.

The SEM images (Figure 8a) clearly show the presence of AgNPs at the edges of the samples (arrows in Figure 8a). This is mainly due to the soaking process to prepare the AgNPs. During soaking, Ag+ ions can diffuse through the hydrogel matrix, but during the reduction step with the ascorbic acid and PVP solution, more AgNPs can be obtained at the edges of the sample, where contact with ascorbic acid is favored.

To ensure that the white areas in Figure 8 were silver nanoparticles, EDX analysis was performed. Table 2 gives the EDX analysis results based on the percentage of atomic composition.

Thus, the white area (Z2) was predominantly composed of silver nanoparticles, while the gray area (Z1) was composed mainly of polymers, as the content of carbon and fluor is higher in the gray zones than in the white ones.

Knowing the areas in which the AgNPs were, further SEM images were taken to verify the particle morphology as well as the morphology of the whole sample. Figure 9 shows the images taken from the hydrogel samples.

As can be observed, the nanoparticles are mainly agglomerated and have a distinct morphology from the other parts of the sample.

Moreover, by the pictures in Figure 9, it was possible to identify that the PVDF, as a hydrophobic polymer, forms big round clusters that are involved by the polymeric matrix that does not allow the PVDF to be released from the sample. Inside the big clusters of PVDF, the small balls represent the particles formed by the polymeric chains of this PVDF polymer.

Therefore, it could be detected that the insertion of AgNPs did not enhance the concentration of the β-phase, as the particles could not enter the PVDF clusters.

Based on that, it can be suggested that the hydrogels with nanoparticles generated a detectable piezoelectric response due to the nanoparticles’ electrical conduction. Adding noble metallic nanoparticles such as silver, gold, or platinum can improve the sensitivity of piezoelectric sensors [32,33]. Moreover, the mechanical stress input induces the material to form internal electrical charges through the piezoelectric effect, primarily associated with the β-phase of PVDF. When the hydrogel is compressed, the asymmetric molecular structure of the β-phase PVDF chains undergoes deformation, leading to a realignment of dipoles and generation of electric potential across the material. This charge separation is enhanced by the presence of conductive AgNPs, which facilitate the movement and collection of charges, effectively converting mechanical energy into an electrical signal. This synergistic interaction between piezoelectric polymer domains and conductive pathways allows the hydrogel to act as a sensitive strain-responsive material [23,24,34].

### 2.4. Three-Dimensional Printing Results

Figure 10 illustrates the fabrication of a hydrogel printed with PVA/PVDF (6%) ink. Due to the fabrication process (DMF removal, freeze-drying, soaking in AgNO_3_, and ascorbic acid solution) the printed hydrogel shrinks slightly but the honeycomb structure is still visible. Samples prepared by direct ink writing confirm good rheological properties of the ink and good manufacturing to consider the development of small sensors with controlled geometry. A brownish color appears at the end of the process, indicating the presence of AgNPs.

As previously demonstrated, PVDF6@AgNP samples exhibit a piezoelectric response. Piezoelectric experiments were conducted on 3D printed samples subjected to various compressive loads. Figure 11 illustrates the piezoelectric responses of 3D printed samples that subjected the entire manufacturing process to different compression cycles. In Figure 11, three different piezoelectric responses can be clearly identified under low, medium, and high compressive loads. These applied loads were selected because they are associated with everyday movements. This represents a light contact or push with a single finger (25 N) up to a moderate push with one hand (100 N) or a strong push with both hands or sport contacts (200 N) [35,36]. Samples that are 3D printed therefore do not exhibit a linear piezoelectric response under mechanical compression. The voltage variation obtained is lower than that obtained previously (Figure 5), which can be explained by the thinner thickness of the 3D-printed sample (2 mm) instead of 8 mm for the conventional sample. A thinner sample has less stimulable material (PVDF and AgNPs) to generate polarization and obtain a large voltage variation. The polarization of the sample, allowing a piezoelectric response to be obtained, is therefore favored by strong compression. The developed hydrogel therefore has different electrical responses to mechanical stress, giving it promising strain sensor properties.

## 3. Conclusions

This study successfully demonstrated the formulation, characterization, and 3D printing of biocompatible piezoelectric hydrogels for use as strain sensors. By incorporating PVA, KC, PVDF, and AgNPs, a composite hydrogel ink was developed with rheological properties suitable for direct ink writing. The ink exhibited shear-thinning and thixotropic recovery properties, essential for controlled extrusion. The final structured hydrogels retained high water absorption and mechanical stability. Through the study of different approaches to inserting the silver nanoparticles in the hydrogels and considering the various concentrations of some raw materials, a procedure was established to form the final strain sensor.

Among the notable results, the presence of AgNPs was shown to be crucial for the piezoelectric response of the hydrogel sensors. Although PVDF is known for its piezoelectricity, its contribution alone was insufficient in the tested formulations. AgNPs significantly improved conductivity and detection performance, although they did not influence the formation of the PVDF β phase, as confirmed by FTIR and XRD analyses. Electron microscopy (SEM) and EDX also revealed the distribution and integration of AgNPs within the hydrogel matrix.

Based on all the generated data, the best ink formulation was set as 5% PVA, 0.94% KC, and 6% (*w*/*w*) PVDF, using as solvents deionized water and DMF in the ratio 7:3 (*v*:*v*). The silver nanoparticles should be produced in soaking steps. Improvements can be made concerning the optimization of the developed procedure, such as verifying if the soaking steps can be executed in a shorter time or optimizing the printing conditions. A reproducible 3D printing protocol was established, enabling the fabrication of sensors with controlled geometry and sensitivity to mechanical stress.

## 4. Materials and Methods

### 4.1. Materials

Polyvinyl alcohol (PVA) (Mowiol 20–98, molecular weight 125,000 g·mol^−1^ and hydrolysis content 98%), and poly(vinylidene fluoride) (PVDF) were provided by Merck (Darmstadt, Germany) and ThermoFisher Scientific (Waltham, MA, USA), respectively. The κ-Carrageenan (KC) (Satiagel UHD 200, batch 01922BIPBA) was a donation from Cargill (Mechelen, Belgium). N,N-dimethylformamide (DMF), L-ascorbic acid and Polyvinylpyrrolidone (PVP) were provided by Merck. Silver nitrate (VWR chemicals) was used to obtain silver nanoparticles (AgNPs). All the reagents were used as received.

### 4.2. Preparation of the Structured Hydrogels

To prepare the hydrogels, PVA was added in deionized water and heated at 90 °C under vigorous stirring until complete dissolution. The temperature was reduced to 70 °C and KC was added under stirring until dissolution. Then, DMF was added under stirring for homogenization. PVDF was added and stirred vigorously until complete dissolution. The concentrations of PVA, KC, and PVDF in these samples were 5% wt, 0.9% wt, and 6% wt (8% wt or 10% wt), respectively. The proportion water:DMF was kept at 7:3 *v*/*v*. Finally, 15 g of hydrogel was poured into a plastic mold (6.0 × 4.0 cm) and allowed to stand until reaching room temperature.

The hydrogels in the mold were subjected to three freeze–thawing cycles to increase the hydrogel strength. Freezing steps took place at −25 °C for at least 3 h and thawing cycles took place at room temperature for 1.5 h. After the completion of these cycles, the samples were placed at −25 °C for at least 3 h before going through lyophilization. The lyophilization was performed at −85 °C at 0.1 mbar for 24 h. When the lyophilization was over, the samples were placed in 200 mL of deionized water for 24 h, to remove the DMF present in the product and to reach the swollen state by diffusion. The water was changed after 4 h and 8 h, respectively.

The AgNPs were formed by reducing silver ions present in an AgNO_3_ solution using an ascorbic acid-PVP solution [37]. The AgNPs were produced after hydrogel formation and lyophilization. In this case, subsequently the DMF removal step, the structured hydrogels were placed in AgNO_3_ solution for 8 h and after this period, this solution was changed to an ascorbic acid-PVP solution, in which the sample stayed for 24 h. In this way, the nanoparticles were formed after two soaking steps. The manufacturing process is summarized Figure 12.

The term “ink” is used to attribute the hydrogel before the freeze–thawing cycles. The term “structured hydrogels” refers to the inks after the freeze–thawing cycles.

The code associated with the produced samples in this stage is shown in Table 3.

### 4.3. Three-Dimensional Printing Protocol

Three-dimensional printing protocols have been developed with a Bio X printer (Cellink, Gothenburg, Sweden). The nozzle size used was 22 G. The inks were printed at 20 °C under a pressure of 65 kPa and a speed of printing equal to 3.5 mm·s^−1^. Pre- and post-flow times of 100 ms were added to optimize the printing parameters with the rheological behavior of the inks. The printbed was kept at room temperature. A 2 cm diameter disk with 3 mm of thickness and a 25% infill with honeycomb structure have been printed with the PVA/PVDF 6% wt.

### 4.4. Characterization of Inks and Hydrogels

#### 4.4.1. Water Uptake of Hydrogels: Swelling Ratio Analysis

For the samples without AgNPs, each sample was cut into three pieces, each of them with approximately 0.4 g. The initial weight was recorded and then these samples were placed in a beaker with 50 mL of deionized water. At certain intervals, the samples were taken from the beaker and carefully dried to take the excess water present on the surface of the materials. In the sequence, their weight was recorded and they were placed back again in the same beaker.

The intervals in which the mass was recorded were: 0, 0:30, 1:00, 1:30, 2:00, 3:00, 4:00, 5:00, 6:00, 8:00, and 24:00 h.

The same procedure was applied to the samples with the nanoparticles, but the initial mass was approximately 0.2 g and the amount of water in the beaker was 25 mL.

The swelling ratio (*SR*) was calculated as described in Equation (3.3) [7]. In Equation (1), *M*_0_ represents the initially recorded mass of the sample, and *M_t_* represents the mass recorded in a specific time “*t*”.(1)$$SR\;(\%)=\frac{\left(M_{t}-M_{0}\right)\times 100}{M_{0}}$$

#### 4.4.2. Rheology Measurements

The rheometer used to characterize the different inks and structured hydrogels was a Kinexus Pro, from Netzsch (Selb, Germany). The ink samples needed to be heated before the analysis as well as the rheometer’s plate to ensure that the whole geometry would be covered correctly. For the different inks, the temperature was 45 °C without AgNPs and 70 °C with AgNPs to allow them to become fluid. The temperature was only reduced to 25 °C when the geometry was already in contact with the sample and the trim was already executed. Then, the temperature of the rheometer’s plate was reduced to 25 °C. When this temperature was reached, the samples stayed for 15 min at this temperature to guarantee that the materials would be at 25 °C when the measurement started. For structured hydrogels, the temperature was directly maintained at 25 °C for experiments.

The rheological performance of the inks was evaluated by the following experiments:Flow test

In the flow test, the applied shear rate was in the range of 0.01–1000 s^−1^. The gap used was 0.15 mm and the geometry adopted was the conical plate with a 4° angle and 40 mm of diameter. This geometry was selected for this experiment because using the conical geometry combined with the parallel plate base, the shear rate is constant in the whole sample, not varying as a function of the radius, so the absolute value of viscosity is measured with more accuracy. All measurements were performed at 25 °C.

Shear stress sweep

The shear stress sweep experiment was performed using the frequency of 1 Hz, with the shear stress varying from 0.01 to 1000 Pa. The gap was 1 mm and the geometry used was the parallel plate with 20 mm of diameter.

Amplitude sweep

For amplitude strain sweep measurements, the strain range used was from 0.01 to 1000%, the measurement gap was set at 1 mm, and the frequency was 1 Hz. The geometry adopted was the parallel plate with 20 mm of diameter.

Frequency sweep

For the frequency sweep measurements, the frequency range used was from 10 to 0.01 Hz, the measurement gap was set at 1 mm, and the geometry applied was the parallel plate with 20 mm of diameter. The values of amplitude strain adopted changed between the inks, as they depend on the ink’s linear viscoelastic region. It is necessary to perform this trial in the linear viscoelastic region of the materials, as in this region the elastic and viscous moduli are independent of the strain amplitude, hence the material’s structure is not destroyed, as happens in the non-linear viscoelastic region [38]. For the different inks, the strain value used for the frequency sweep experiment was 1%.

Thixotropy loop

The thixotropy loop measurement was executed using the parallel plate with 20 mm of diameter as the geometry. The low level of stress applied was set at 1 Pa, and the high level was set at 350 Pa, as at this level all inks were out of their respective linear viscoelastic region.

The inks with different concentrations of PVA with and without AgNPs had their rheological performance assessed by amplitude sweep and frequency sweep measurements.

The rheological performance of the structured hydrogels was evaluated by the following experiments:Amplitude strain sweep measurements

For these measurements, the gap was set as the height at which the contact of the geometry with the hydrogel has a force of 0.5 N, to standardize that all samples would be under the same conditions without forcing their network. The gap values are exhibited in Table 4.

In addition to the gap values, the other conditions applied were

Geometry: Parallel plate with 20 mm diameterTemperature: 25 °CFrequency: 1 HzShear strain range: 0.01–1000%

The samples were cut in a circular shape with 20 mm diameter to be analyzed.

#### 4.4.3. Infrared Spectroscopy

IR measurements were executed using a Vertex 70 spectrometer coupled with the PMA 50 accessory, both from Bruker (Billerica, MA, USA). The scanned region was from 400 to 4000 cm^−1^. The number of scans was 32 and the resolution was 4 cm^−1^. Attenuated Total Reflectance (ATR) was the sampling technique used for these analyses.

#### 4.4.4. X-Ray Diffraction

X-ray diffraction measurements were performed using the device D8 Discover (Bruker). The angle (2θ) range adopted was from 10 to 80°. The energy of the radiation was 40 kV and the tube current was 40 mA. The anode was copper (Cu). To execute these measurements, a slit with a 2 mm size was used. The usage of this slit allowed the beam to focus in smaller areas of the sample. The samples were analyzed in the dry state.

#### 4.4.5. Scanning Electron Microscopy Coupled with Energy-Dispersive X-Ray Analysis

The device used was an electron microscope TESCAN MIRA3 (Warrendale, PA, USA). For the EDX measurements, the detector was a Silicon Drift Detector 30 mm^2^ SamX (Warrendale, PA, USA). The energy used to generate images and for the EDX was 7 kV. SEM images were performed with freeze-dried samples (Xerogel samples). The samples were recovered with a carbon layer of 8 nm. This was conducted so that the samples could have linear electrical conductivity on the surface, which is necessary to perform these analyses.

#### 4.4.6. Assessment of Piezoelectricity and Resistance Measurements

Piezoelectrical measurements

For the piezoelectric measurements, the samples were first used in the dry state. This approach was used to ensure that the piezoelectricity measured is related to the components used in the hydrogel formulation and to avoid water interference [39]. Then, new probes were cut in a circular shape with a 2 cm diameter and placed in a beaker with 20 mL of deionized water for 30 min. Subsequently, the samples were removed from the water and placed in a plastic Petri dish to dry at room temperature for 48 h, to lose the excess of water for the piezoelectric measurements. Dry samples were also cut into a disk shape with the same dimension.

To measure piezoelectricity, the output voltage of the hydrogels was checked under compression cycles. The compression cycles were executed by the compression force tester F505, from MARK-10 (Copiague, NY, USA). The output voltage was measured by the Sourcemeter 2450, from Keithley (Beaverton, OR, USA). The samples were cut in a disk shape with a 2 cm diameter and 8 mm of thickness.

The plates used for compression measurements had a 5 cm diameter. The setup used was an adaptation from the setup used by Y Wang et al. [40]. The plates were covered with insulating tape to ensure that the materials would be electrically isolated and that the only resistance measured would be associated with the hydrogel sample. On each plate, one copper tape strand with dimensions 6 cm × 2 cm, was placed on the insulating tape; 3.5 cm of the tape was exposed on the insulating tape, while the rest was used to connect the connectors from the multimeter. Each sample was submitted to 30 cycles of compression, with the maximum load applied as 50 N. The dwell time was 5 s and the speed of the head to go up and down was set at 10 mm/min.

Resistance measurements

These measurements were made using the same setup explained for piezoelectricity measurements. However, instead of using compression cycles, the upper plate was moved to the height at which it was touching the whole upper surface of the hydrogel. Thus, the measurement was static. Triplicates were performed for each sample, each probe from a different part of the sample and at least three different samples per formulation were characterized.

Moreover, as was evaluated for the output resistance of the materials, an input voltage of 2 V was applied for 1 min.

The samples used for these essays were in a dry state and had the same dimensions described for piezoelectricity characterizations.

## Figures and Tables

**Figure 1 gels-11-00474-f001:**
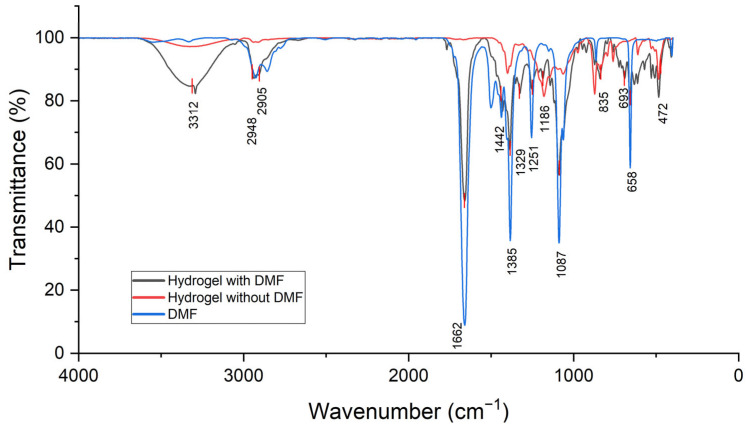
IR spectra to verify the efficiency of the DMF removal step.

**Figure 2 gels-11-00474-f002:**
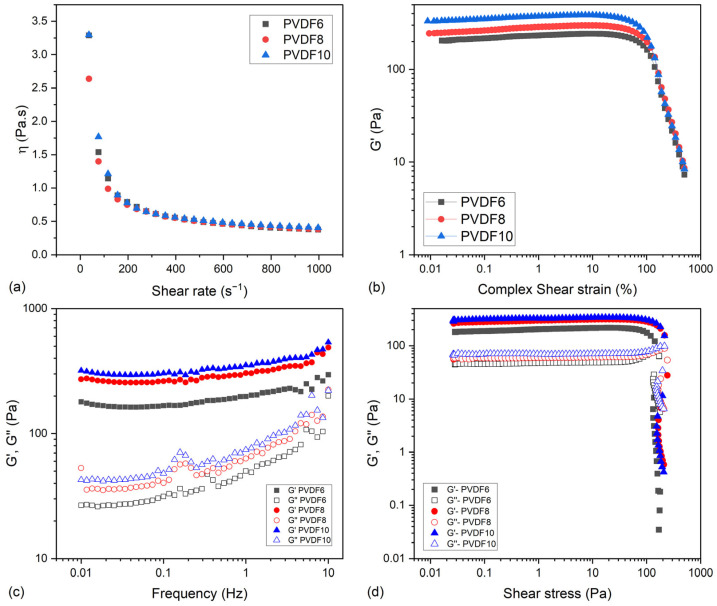
Rheological characterizations of PVA/KC inks with different concentrations of PVDF. (**a**) Viscosity measurements, (**b**) amplitude strain sweep measurements, (**c**) frequency sweep measurements, and (**d**) shear stress sweep measurements.

**Figure 3 gels-11-00474-f003:**
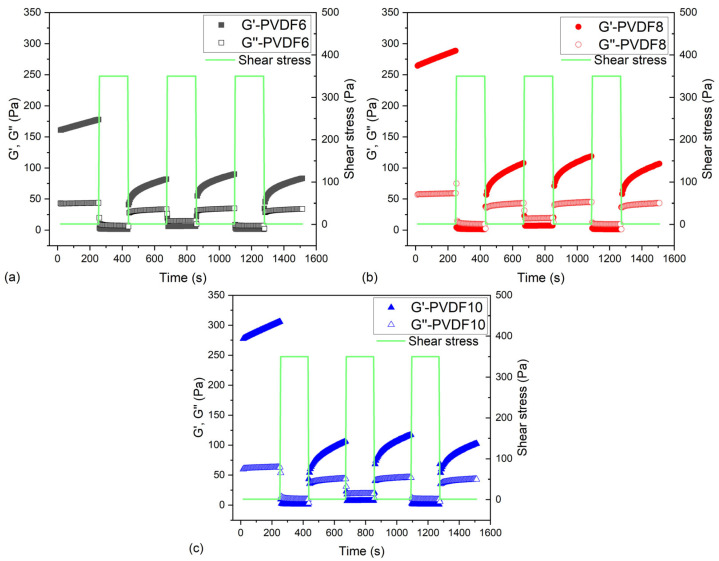
Thixotropy loop measurement for the samples with different PVDF concentrations. (**a**–**c**) for 6, 8, and 10% PVDF content, respectively.

**Figure 4 gels-11-00474-f004:**
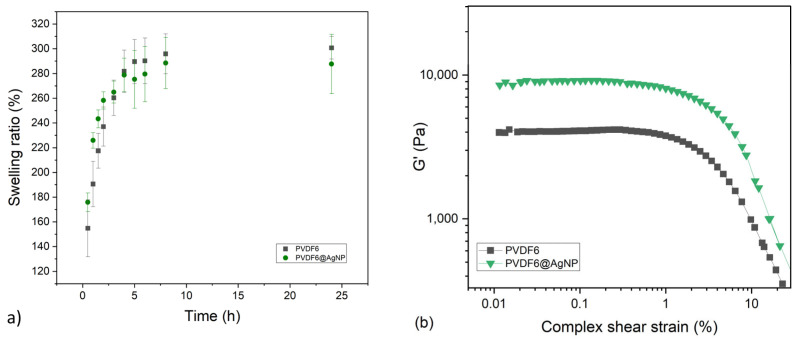
(**a**) Swelling ratio measurement and (**b**) amplitude strain sweep measurement for structured hydrogels with and without AgNPs.

**Figure 5 gels-11-00474-f005:**
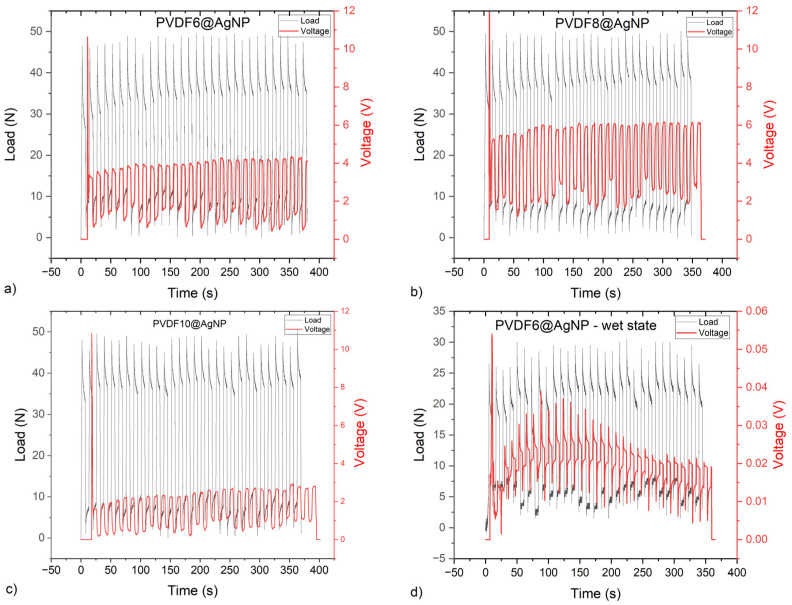
Piezoelectric response of the samples with AgNPs (**a**–**c**) in the dry state and (**d**) in a wet state.

**Figure 6 gels-11-00474-f006:**
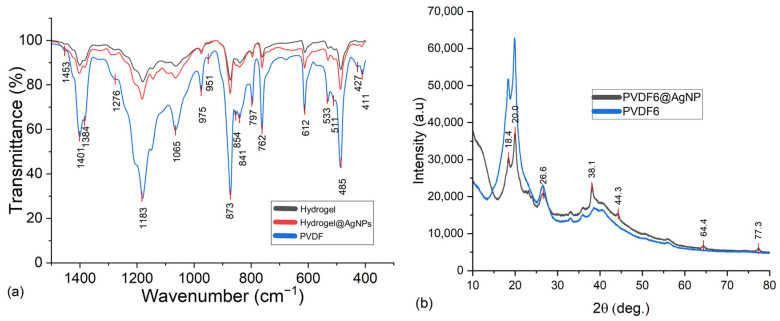
(**a**) Comparison of IR spectra and (**b**) XRD analysis for a hydrogel to identify the PVDF polymorphic phases and AgNPs.

**Figure 7 gels-11-00474-f007:**
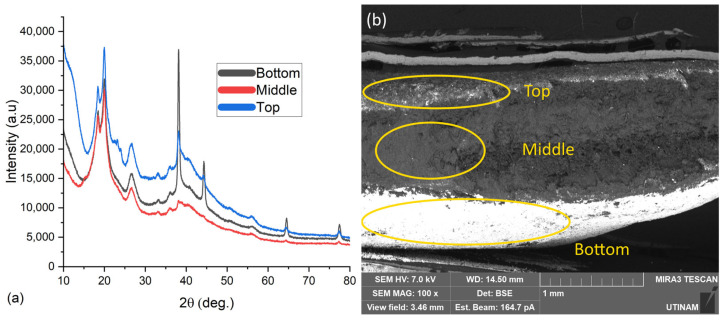
(**a**) XRD analysis in three different regions (top, bottom, and middle) of the same hydrogel with AgNPs. (**b**) SEM picture showing the regions referenced in the XRD analysis.

**Figure 8 gels-11-00474-f008:**
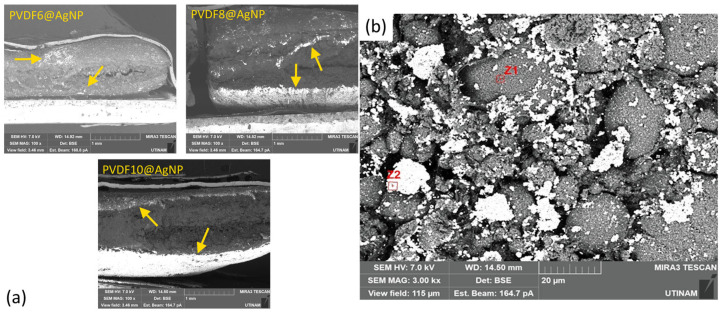
SEM images of (**a**) the hydrogels to identify the AgNPs distribution in the sample. The arrows indicate the main regions in which AgNPs were present and (**b**) the selected area for EDX analysis.

**Figure 9 gels-11-00474-f009:**
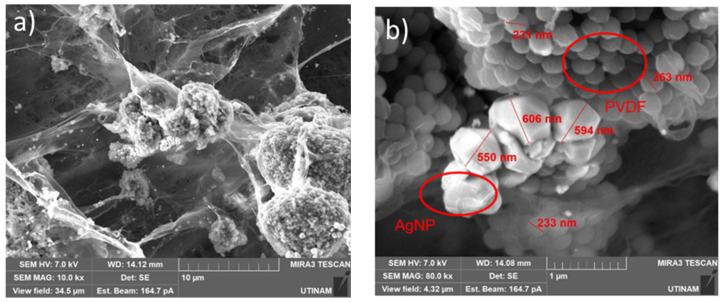
(**a**) SEM image of the morphology of hydrogel. (**b**) SEM image of AgNPs and PVDF clusters.

**Figure 10 gels-11-00474-f010:**
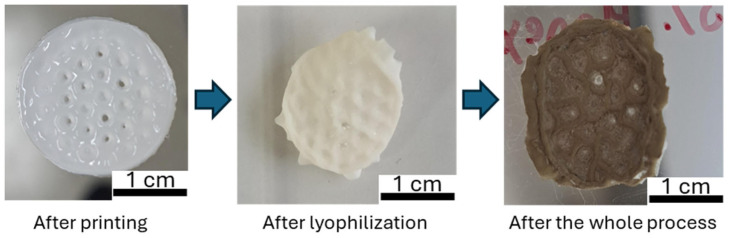
Picture of 3D printing sample during the manufacturing process.

**Figure 11 gels-11-00474-f011:**
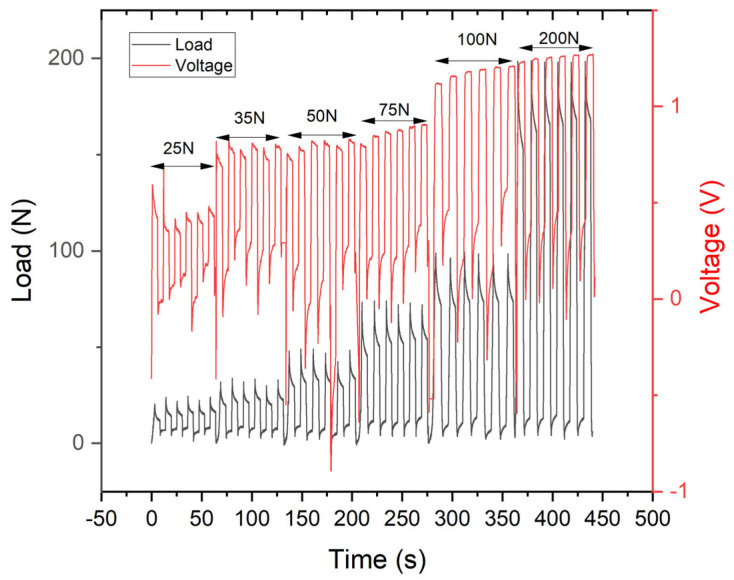
Piezoelectric response of the 3D printing sample in a dry state.

**Figure 12 gels-11-00474-f012:**
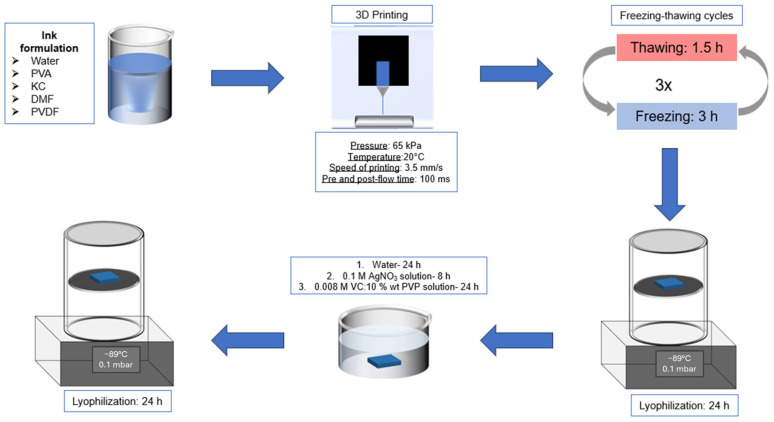
Schematic protocol to produce structured hydrogel with AgNPs for strain sensors evaluation.

**Table 1 gels-11-00474-t001:** Average resistance values of the hydrogels in the dry and wet state.

Sample	Average Resistance (GΩ)	Average Resistance (kΩ)
	Dry State	Wet State
PVDF6	58 ± 18	227 ± 44.2
PVDF8	127 ± 19	120 ± 25.1
PVDF10	29 ± 5	344 ± 56.4
PVDF6@AgNP	1.7 ± 0.8	1.29 ± 0.65
PVDF8@AgNP	1.2 ± 1.1	1.34 ± 0.54
PVDF10@AgNP	1.8 ± 1.2	1.38 ± 0.37

**Table 2 gels-11-00474-t002:** EDX results for the different zones of the hydrogel.

Zone	Atomic Composition (%)			
	C	O	F	Ag
Z1	69.8370	03.1950	26.3426	00.6253
Z2	12.5521	00.0001	00.3055	87.1423

**Table 3 gels-11-00474-t003:** Code associated with the composition of each sample.

Final Sample Composition	Sample’s Code
PVA, KC, PVDF (6% wt)	PVDF6
PVA, KC, PVDF (8% wt)	PVDF8
PVA, KC, PVDF (10% wt)	PVDF10
PVA, KC, PVDF (6% wt), AgNPs	PVDF6@AgNP
PVA, KC, PVDF (8% wt), AgNPs	PVDF8@AgNP
PVA, KC, PVDF (10% wt), AgNPs	PVDF10@AgNP

**Table 4 gels-11-00474-t004:** Measurement gap values used for the amplitude sweep measurements of the structured hydrogels.

Sample	Measurement Gap (mm)
PVDF6	1.84
PVDF8	1.26
PVDF10	1.39
PVDF6@AgNP	1.06
PVDF8@AgNP	1.48
PVDF10@AgNP	1.64

## Data Availability

The data presented in this study are available in the article.

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
