# Peer review of "Elaboration of Conductive Hydrogels by 3D Printer for the Development of Strain Sensors"

_gels, 2025, doi:10.3390/gels11070474_

Round 1

Reviewer 1 Report

Comments and Suggestions for Authors

see attached file

Author Response

Thank you very much for taking the time to review this manuscript. Your comments definitively help to improve this manuscript by giving it more clarity and precision. Please find the detailed responses below and the corresponding revisions/corrections
highlighted/in track changes in the re submitted files.

Reviewer 2 Report

Comments and Suggestions for Authors

This study presents the formulation and 3D printing of biocompatible, piezoelectric hydrogels composed of PVA, κ-carrageenan, PVDF, and AgNPs, optimized for direct ink writing (DIW) and strain sensing applications. The authors systematically evaluated the rheological, mechanical, and electrical properties of the inks and structured hydrogels, demonstrating that silver nanoparticles are essential for piezoelectric performance, even though they do not enhance the β-phase of PVDF.
Here are my comments:
The manuscript attributes improved piezoelectric response to AgNPs' conductivity, but this remains speculative. There is no quantitative model or detailed mechanistic discussion of charge polarization or interfacial effects.
The observed edge-accumulation of AgNPs, due to the soaking/reduction method, introduces heterogeneity. The implications for device uniformity and reproducibility are not explored experimentally.
There's no evaluation of sensor fatigue, stability over time, or under repeated mechanical stress.
This paper lacks reviewing recent studies in introdution, like Machine learning-enhanced soft robotic system inspired by rectal functions to investigate fecal incontinence; Design, modeling, and characteristics of ring-shaped robot actuated by functional fluid.
The piezoelectric signal is presented in volts without load normalization or signal-to-noise evaluation. There's also no benchmark comparison with state-of-the-art flexible sensors.
Minor grammatical errors throughout (e.g., “hydrogel that passed through the DMF removal” should be “hydrogel after DMF removal”).
Figures lack units or axes labels in some cases, making interpretation harder.
The introduction could better contextualize the gap in current literature (e.g., how current PVA/PVDF or AgNP-only systems fail).
Repetition in the abstract and introduction (e.g., repeated statements of project aim).

Author Response

Please find comments in the attached file

Reviewer 3 Report

Comments and Suggestions for Authors

In this paper, the authors investigate the properties of hydrogels produced by 3D printing from an ink composed of polyvinyl alcohol, κ-carrageenan, polyvinylidene fluoride, and silver nanoparticles. The authors' 3D printing materials and hydrogels are noteworthy because they provide experimental results that form the basis for utilizing 3D printing technology in the creation of hydrogel materials.

The authors' article is well-structured, the references are well-chosen, and the examples of their work are presented with plenty of specificity, but contain the following minor issues that could be improved.

  1. Title: Since the title states "biocompatible conductive hydrogels," the authors should include the results of their biocompatibility evaluation.
  2. The authors should describe the required properties of ink for 3D printing and the resulting hydrogel and compare and examine whether these properties were achieved in comparison with those of previous 3D printing materials.
  3. By clearly explaining the research concept behind their choice of components for this composite material, the authors' research results will become clearer.
  4. 2.1 Efficiency of the DMF removal step: The use of DMF in the experimental system is assumed, but there is no description of the background for using DMF. The authors should explain the type of ink they created and describe the necessity of removing DMF. Additionally, the authors should specify the official name of DMF in this section.
  5. Figures 1 and 6: Rather than simply listing raw data, the authors should scientifically correct the significant digits of the numerical data, considering the measurement conditions, which would enhance the reliability of their measurement results.
  6. The authors should provide a detailed description of the ink's gel state characterization. They should also describe in detail which experimental data was used to determine that it is a hydrogel.
  7. The authors should describe the particle size distribution of the obtained silver nanoparticles and explain how this particle size affects their research results.
  8. Figure 9: a) SEM image of the morphology of hydrogel. b) SEM image of AgNPs and VDF clusters.: Did the authors observe the SEM image of the hydrogel? Xerogel?
  9. Why did the authors not use 3D printing technology to create objects of various shapes, which is one of the characteristics of this technology, in this study?
  10. There is no information on the grade (molecular weight or saponification degree) of the reagents used. The same applies to κ-carrageenan. The authors should describe these details.

Author Response

(The authors gave the same response as above.)

Round 2

Reviewer 2 Report

Comments and Suggestions for Authors

ok

Reviewer 3 Report

Comments and Suggestions for Authors

As shown in the revised manuscript, some issues suggested by the reviewer were resolved.

The reviewer believes the authors' findings will contribute to advancing the science and chemistry of gel-related fields.

The reviewer recommends accepting the revised manuscript for publication in “Gels."